# The Effects of El Niño-South Oscillation on the Winter Haze Pollution of China

Shuyun Zhao[1,2], Hua Zhang[1,2], Bing Xie[1,2]

[1]Laboratory for Climate Studies, National Climate Center, China Meteorological Administration, Beijing, China

[2]Collaborative Innovation Center on Forecast and Evaluation of Meteorological Disasters, Nanjing University of Information Science & Technology, Nanjing, China

*Correspondence to*: Hua Zhang (huazhang@cma.cn)

**Abstract.** It has been reported in previous studies that El Niño-South Oscillation (ENSO) influenced not only the summer monsoon, but also the winter monsoon over East Asia. This contains some clues that ENSO may affect the winter haze pollution of China, which has become a serious problem in recent decades, through influencing the winter climate of East Asia. In this work, we explored the effects of ENSO on the winter (from December to February) haze pollution of China statistically and numerically. Statistical results revealed that the haze days of southern China tended to be less (more) than normal in El Niño (La Niña) winter; whereas the relationships between the winter haze days of northern and eastern China and ENSO were not significant. Results from numerical simulations also showed that ENSO influenced the winter atmospheric contents of anthropogenic aerosols over southern China more obviously than it did over northern and eastern China. Under the emission level of aerosols for the year 2010, the winter atmospheric contents of anthropogenic aerosols over southern China were generally more (less) than normal in El Niño (La Niña) winter. It was because that the transports of aerosols from South and Southeast Asia to southern China were enhanced (weakened), which masked the better (worse) scavenging conditions for aerosols in El Niño (La Niña) winter. The frequency distribution of the simulated daily surface concentrations of aerosols over southern China indicated that the region tended to have less clean and moderate (heavy) haze days, but more heavy (moderate) haze days in El Niño (La Niña) winter.

## 1 Introduction

Haze pollution, especially in winter, has become a very serious problem for China in recent decades (Ding and Liu, 2014; Tao et al., 2016). For example, in January 2013, most parts of central and eastern China experienced an extremely heavy and

persistent haze pollution (Tao et al., 2014; Mu and Zhang, 2014; Zhang et al., 2014; Wang et al., 2014a; b; Zou et al., 2017). In the last decade, haze pollution in winter has received wide concerns from the scientific community, the government of China, and the public.

Haze pollution is a phenomenon mainly caused by human-emitted pollutants under stagnant meteorological conditions. The increase in anthropogenic emissions of aerosols and their precursors in recent decades has been the main reason for the worsening air qualities in China (Cao et al., 2007; Zhang et al., 2012; Zhu et al., 2012). In addition to the increase in anthropogenic emissions of aerosols and their precursors, climate change caused by anthropogenic and/or natural forcings also exerted great influences on the haze pollution in China, especially through changing the strength of the East Asian Monsoon (Zhang et al., 2010; 2014; Liu et al., 2011; Yan et al., 2011; Chin, 2012; Zhu et al., 2012; Mu and Zhang, 2014; Chen and Wang, 2015; Li et al., 2016a; b; Cai et al., 2017). Studies generally showed that the wintertime haze days across central and eastern China had a close negative relationship with the strength of the East Asian Winter Monsoon (EAWM) (Zhang et al., 2014; Mu and Zhang, 2014; Chen and Wang et al., 2015; Li et al., 2016a; Cai et al., 2017). In summer, the increase in surface aerosol concentration and optical depth over eastern and northern China correlated with the weakening of the East Asian Summer Monsoon (EASM) (Zhang et al., 2010; Yan et al., 2011; Zhu et al., 2012). Whereas Yang et al. (2014) found that the gaseous pollutant of surface $O_3$ over China had a positive relationship with the EASM.

Wang et al. (2015) and Zou et al. (2017) revealed that the increasing winter haze pollution in eastern China in recent years was related to the decreasing Arctic sea ice in preceding autumn. Many studies suggested that under global warming, the future climate would be more stagnant and the weather conditions conducive to severe haze in eastern China would be more frequent (Jacob and Winner, 2009; Wang et al., 2015; Cai et al., 2017). But Jacob and Winner (2009) also pointed out that the effects of future climate change on particulate matter (PM) were complicated, as the projection of precipitation, wildfires, atmospheric chemistry, and natural emissions of aerosols by models were still in need of improvement.

ENSO is the dominant changing mode of the tropical sea surface temperature (SST) on interannual scale (Rasmusson and Carpenter, 1982), of which the climatic effects are global (Bjerknes, 1972; Huang and Wu, 1989; Zhang et al., 1999; Lau and Nath, 2003; Zhai et al., 2016, etc.). ENSO not only influences the EASM (Chang et al., 2000; Li et al., 2007; Zhao et al., 2017, etc.), but also influences the EAWM, especially over low latitudes (Chen et al., 2013; He and Wang, 2013; Kim et al., 2016,

etc.), which indicates that ENSO may affect the haze pollution over East Asia through influencing the monsoon circulation. Wu et al. (2013) found that the aerosol variations over the Maritime Continent and western North Pacific presented a biennial feature, which could be attributed to the impacts of ENSO. From the results of Wu et al. (2013), it seemed that ENSO only influenced the aerosols over eastern China (30°–40°N, 110°–120°E) around October in El Niño or La Niña developing years and July in El Niño or La Niña decaying years. Gao and Li (2015) revealed statistically and through case analyses that El Niño (La Niña) events were more likely to bring more (less) haze days in eastern China (25°–35°N, 105°–122.5°E). Feng et al. (2016) simulated the influence of the 1994/1995 El Niño Modoki event on the aerosol concentrations over southern China (20°–35°N, 105°–120°E), and found that the aerosol concentrations increased during the mature phase (in boreal winter) of the event. Feng et al. (2017) simulated the influences of the 1998/1999 and 2000/2001 La Niña Modoki events on the aerosol concentrations over eastern China (105°–120°E), and found that the geographical distributions of aerosols over eastern China during the winters of the two events were opposite with each other. Most previous studies discussed the effects of ENSO on the winter haze pollution of China based on observation or case simulation separately. In this study, we try to explore the effects of ENSO on the winter haze pollution of China statistically and numerically. ENSO is a recurring climate pattern, and many climate centers of the world monitor it systematically and use it in climate prediction extensively. Therefore, exploring the effects of ENSO on the winter haze pollution in China may provide useful information in the prediction of haze for the country.

The article is organized as follows: methodology is given in section 2, including data and model introduction; results including statistical and model results, are presented in section 3, followed by conclusions and discussions in section 4.

## 2 Methodology

### 2.1 Data used in statistical analysis

We firstly analyzed the relationship between the winter haze days of China and global SST in section 3. The monthly haze days from the data set for haze project version 1.0 of the National Meteorological Information Center, China Meteorological Administration (CMA) were used. The CMA defines haze using visibility (<10 km) and relative humidity (<80%) (Tao et al., 2014). The time-span of the data set is January 1954~July 2014, but only the data after 1960 were actually used in this study, as the data set has steadily included more than 2000 stations since 1960. A merged monthly SST data (Hurrell, et al., 2008)

formed by the SST data of the Hadley Centre and the National Oceanic and Atmospheric Administration (NOAA) was used in the correlation analysis. The SST data together with a merged sea ice (SI) data of the same sources were used in the following numerical simulation. The SST and SI data are both from 1870 to 2012, with a horizontal resolution of 1°×1°.

## 2.2 Model description and experimental set-up

The aerosol-climate coupled model BCC_AGCM2.0_CUACE/Aero (Zhang et al., 2012) of the National Climate Center (NCC), CMA, was used in the numerical study of the effects of ENSO on the atmospheric contents of aerosols. The coupled model composes of the NCC/CMA climate model (BCC_AGCM2.0, Wu et al., 2010) and the CMA Unified Atmospheric Chemistry Environment/Aerosol model (CUACE/Aero, Gong et al., 2002, 2003). The coupled model employs a horizontal T42 spectral resolution (about 2.8°×2.8°) and a hybrid vertical coordinate with 26 levels, the top of which is located at about 2.9 hPa. Five

types of aerosols (including their emissions, gaseous chemistry, transports, coagulations, and removals): sulfate (SF), black carbon (BC), organic carbon (OC), dust, and sea salt are considered in the model. The emissions of the first three types of aerosols and/or their precursors are prescribed, and the last two types of aerosols are emitted online (Gong et al., 2002). The particle radii of each type of aerosol are divided into 12 size bins from 0.005 to 20.48 μm. All types of aerosols are assumed to be externally mixed with each other. Sulfate, organic carbon, and sea salt are considered to be hygroscopic, and the other

two types of aerosols are considered to be non-hygroscopic. The coupled model has been introduced, evaluated, and used in many studies of the radiative forcings and climatic effects of aerosols (e.g., Zhang et al., 2014; Wang et al., 2015; Zhao et al., 2014; Zhang et al., 2016).

Three groups of experiments were conducted (Table 1), named CLI, EL, and LA, with each group including 20 members by altering initial conditions. To get different initial conditions, a preparation experiment was run firstly with the model's default

setting (Zhao et al., 2014). Three types of files (initial, restart, and history files) of the preparation experiment were output, and the output frequency was set to daily. 20 initial files output from the preparation experiment were then used as the different initial conditions for different ensemble members. The group of CLI used the climatological-mean (from 1981 to 2010) monthly SST and SI, which had been introduced in section 2.1 but interpolated to the model's resolution, as boundary conditions. In the groups of EL and LA, the climatological-mean monthly SST was superposed by El Niño and La Niña SST

perturbations, respectively, and the SI was identical to that in CLI. The El Niño and La Niña SST perturbations were obtained

by scaling a typical ENSO mode (Figure 1a) with the average monthly Niño3.4 indices of 21 El Niño and 18 La Niña events (selected from 1951 to 2015, Figure 1b), respectively. The Niño3.4 indices from January 1951 to date can be downloaded from the website of NCC/CMA (cmdp.ncc-cma.net/download/Monitoring/Index/M_Oce_Er.txt). The typical ENSO mode was obtained through the regression between the monthly Niño3.4 index and the SST field after removing their linear trends. The running period of the group of CLI was from October to the next August, to testify if the model can capture the general features of the circulations of the East Asian winter and summer monsoons. In the testifying process, the geopotential height and wind from the National Centers for Environmental Prediction (NCEP) reanalysis data were used. Whereas the running periods of EL and LA were both from October to the next February. The results in boreal winter (December, January, and February, or DJF for short) of the three groups were used in analysis, allowing prior two months for the atmosphere to response to SST perturbations.

The emission data of SF, BC, OC, and/or their precursors used in all experiments were from the Representative Concentration Pathway 4.5 (RCP 4.5) of the Intergovernmental Panel on Climate Change (IPCC) for the year of 2010. In this study, only the changes in the atmospheric contents of SF, BC, and OC caused by different SST perturbations were analyzed, as the three types of aerosols were mainly emitted by anthropogenic activities and the important components of haze pollutants. In the analyzing process, the median instead of the average of the changes of a specific variable between different groups of experiments (e.g., EL − CLI) was used, as median is more robust and resistant to extreme large or small values that may happen in some ensemble members. And we also marked the grids where the differences of a specific variable between more than 70% pairs of ensemble members had the same sign with the median differences, as a reflection of significance.

## 3 Results

### 3.1 Statistical results

In this section, we firstly presented the geographical distribution of the winter haze days in mainland China over the past about 50 years and since the year of 2000. Then, three typical polluted regions were selected, and the correlations between their respective winter haze days and global SST were analyzed.

It's seen from Figure 2a that Beijing, the southwest part of Hebei, the central and south parts of Shanxi, the central part of Shaanxi, and the north part of Henan suffered winter haze pollution more frequently than elsewhere in China during 1960−

2013, with the largest value of DJF mean monthly haze days among 10~20. The diffusive conditions of these areas are not good because of the influences of the Tai-hang and Qin-ling mountains. Besides the above areas, stations with more than 5~10 DJF mean monthly haze days during 1960−2013 distributed densely in the provinces of Hubei, Hunan, Jiangxi, Zhejiang, Guangxi and Guangdong. Compared with 1960−2013, winter haze pollution in mainland China generally became more frequent during 2000−2013 (Figure 2b). For example, in the Yangtze River Delta and Pearl River Delta, stations with 5~10 and 10~20 DJF mean monthly haze days during 2000−2013 were much more than that during 1960−2013. At some stations of Shanxi province, the DJF mean monthly haze days during 2000−2013 were even more than 20 (Figure 2b). The locations of the Chinese provinces mentioned above and in the following, can be found in Figure S1 in the supplementary document.

Three representative regions: JJJ (Beijing, Tianjin, and Hebei province; accounting for 179 stations), JZH (Jiangsu and Zhejiang provinces, and Shanghai; accounting for 164 stations), and GG (Guangdong and Guangxi provinces; accounting for 178 stations) were selected to represent northern, eastern, and southern China, respectively. It was seen from Figure 3 that the DJF mean monthly haze days of these three regions were generally less than 3 before the year of 2000, with a small peak around 1980 in northern and eastern China. After the year of 2000, the DJF mean monthly haze days over eastern and southern China grew rapidly. The DJF mean monthly haze days of northern China increased much later than the other two regions after the year of 2000. Actually, the DJF mean monthly haze days over northern China were relatively few until 2012, which was consistent with the result of Chen and Wang (2015).

Considering that the increases in the DJF mean monthly haze days over eastern and southern China increased too abruptly after the year of 2010, only the winter haze days of the three regions from 1960 to 2010 were used in analyzing their relationships with winter SST. In order to remove the inter-decadal variabilities, we applied a 2-8 years band-pass filtering to both the data of haze days and SST. It was seen from Figure 4 that only the winter haze days of southern China had significant negative relationships with the equatorial SST over central and eastern Pacific and central Indian Ocean, and positive relationships with the equatorial SST over western Pacific. The geographical distribution of the correlation coefficients between the winter haze days over southern China and SST was generally an opposite phase of the typical ENSO mode shown in Figure 1a, indicating that southern China tended to suffer more (less) haze days than normal in La Niña (El Niño) winter. This could also be seen from the comparison of the DJF mean monthly haze days between several selected pairs of La Niña

and El Niño winters over southern China (Figure S2). In order to avoid the influences of the variations of the emissions of haze particles and their precursors, each pair of La Niña and El Niño winters were selected with their interval not longer than 2 years.

The relationships between the winter haze days of northern and eastern China and equatorial SST were not significant (Figures 4a and 4b), indicating that ENSO did not influence the winter haze days of these two regions significantly. It's probably because that as a tropical phenomenon, ENSO affects the climate over southern China more directly than that over northern and eastern China, especially in winter when the Western Pacific Subtropical High (WPSH) is generally weaker and located more south than other seasons. Zou et al. (2017) linked the extreme winter haze events over East China Plains (112°–122°E, 30°–41°N, including JJJ and most parts of JZH) to Arctic sea ice loss in the preceding autumn and extensive Eurasia snowfall in early winter. Gao and Li (2015) found that the winter haze days of eastern China had positive relationship with the SST over eastern equatorial Pacific. But the "eastern China" in Gao and Li (2015) included most regions between the Yangtze and Yellow rivers east of 105°E, which was much larger and more west than the representative region of eastern China in this work, and more south and west than the concerned region in Zou et al. (2017).

### 3.2  Model results

### 3.2.1 Winter and summer circulations, and the atmospheric contents of aerosols

First of all, the simulated winter and summer circulations over East Asia in the group of CLI were testified by comparing with NCEP reanalysis data (Figure 5). The model could capture the general features of the winter and summer circulations over East Asia. For example, in winter, the deepening of East Asian Trough (EAT), the overwhelming of northwesterly over the east part of China, and the strengthening of the easterly north of the equator were all depicted by model results (Figure 5a); In summer, the northward shift of WPSH, and the strengthening of the cross-equatorial westerly over the Indian Ocean and Maritime Continent were also generally captured by the model (Figure 5b). But the simulated EAT in winter by the model was weaker and narrower, and located more west than NCEP reanalysis data. In summer, the simulated WPSH was weaker and located more east than NCEP reanalysis data. It seemed that the simulated cross-equatorial flow was stronger than reanalysis data both in winter and summer, which was probably the reason for the weakness of the simulated EAT in winter and WPSH in summer. In the model results of the group of CLI, the stronger cross-equatorial westerly over the Indian Ocean and Maritime

Continent obstructed the westward stretch of the WPSH in summer, resulting in positive precipitation biases over South Asia and Southeast Asia, and negative precipitation biases over southern China (Zhao et al., 2014).

The simulated winter surface concentrations (CONCsur) and loadings of aerosols in the group of CLI (Figure 6) showed that central and eastern China (about east of 105°E) were the most haze polluted regions of the country, in line with the observational distribution of winter haze days shown in Figure 2. The maximum of the winter CONCsur of aerosols centered in Henan province, and was about 20 µg m$^{-3}$. Compared with other observational studies (e.g., Cao et al., 2007; Zhang and Cao, 2015; Cai et al., 2017), the simulated CONCsur of aerosols shown in Figure 6a were underestimated by about 1~2 orders of magnitude. It should be illustrated that the aerosol CONCsur in this study were the aerosol concentrations at the lowest level of the model. As has been introduced in section 2.2, the model used in this study has 26 levels in the vertical hybrid σ-pressure axis, and the mid height of the lowest level is about 50 ~ 70 meters above the surface in China (not shown). Therefore, the aerosol concentrations at the lowest level of the model actually reflect the mean of the aerosol concentrations from the surface to a height of about 100 ~ 140 meters (or maybe even higher) above the surface. This certainly brought about underestimation as to aerosol CONCsur. Another reason for the underestimation was that we excluded all nature-emitted aerosols (dust and sea salt), in order to focus our attentions on haze, as another meteorological disaster occurring frequently in late winter and spring over northern China – sandstorm – has quite different weather conditions with haze. The exclusion of nature-emitted aerosols was also the reason why the northwestern China was much cleaner than it was expected to be.

The maximum of the simulated winter aerosol loadings in China was 21 ~ 25 mg m$^{-2}$, and heavy aerosol loadings (≥18 mg m$^{-2}$) in China located east of 105°E, and between the Yellow and Yangtze rivers (Figure 6b). The median of the 10 models that participated in the Aerosol Comparisons between Observations and Models (AeroCom, http://aerocom.met.no/cgi-bin/aerocom/surfobs_annualrs.pl) showed that the maximums of the loadings of SF, BC, and OC in January 2000 in China were 15 ~ 30, 2.5 ~ 5, and more than 10 mg m$^{-2}$, respectively. The model used in this study is also a member of AeroCom. Zhao et al. (2014) have compared the simulated loadings of five kinds of aerosols (SF, BC, OC, sea salt, and dust) by the model with AeroCom median, and found that the model generally simulated the distributions and magnitudes of aerosol loadings

well, though with some underestimations in BC and OC loadings.

### 3.2.2 The effects of ENSO on winter circulation and precipitation

The effects of ENSO on the winter circulation and precipitation over East Asia were discussed firstly, as these meteorological fields determine the transports, diffusions, removals, and consequently the atmospheric contents of aerosols.

Two important features were apparent in the winter anomalous circulation field caused by El Niño (Figure 7a). Firstly, negative and positive anomalous geopotential heights at 500 hPa were seen near Ural Mountains and Lake Baikal, respectively. Secondly, two anomalous anticyclones at 850 hPa were seen over western North Pacific, with one in the Philippine Sea and the other one in the mid latitude. It was expected that the anomalous southwesterly in the northwest of the Philippine Sea anticyclone would bring more water vapor and also more aerosols to southern China, as Indo-China Peninsula and South Asia were both areas with heavy aerosol loadings (Figure 6b). Whereas the anomalous geopotential heights at 500 hPa caused by La Niña were positive and negative near Ural Mountains and Lake Baikal, respectively, and two anomalous cyclones were caused by La Niña over western North Pacific (Figure 7b).

The differences in circulation between El Niño and La Niña winters were shown in Figure 7c. It was seen in Figure 7c that the negative and positive anomalies in 500 hPa geopotential heights caused by El Niño were more obvious than that in Figure 7a over Ural Mountains and Lake Baikal, respectively. And the anomalous anticyclones in the western North Pacific were also more obvious than that in Figure 7a. Wang et al. (2000) found that the anomalous anticyclones over western North Pacific formed in the boreal autumn of a developing El Niño, attained its peak in winter, and persisted into the following spring and early summer. In the sea level pressure field, the land-sea gradient in winter was decreased by El Niño (not shown), indicating that El Niño caused a weakness of the EAWM. The weakness of the EAWM caused by El Niño could also be seen from the weakness of East Asian jet stream and the decrease in EAWM index caused by El Niño (Figure S3). The EAWM index used in Figure S3 was the one defined by Li and Yang (2010). It has been suggested in previous studies that the weakness of East Asian jet stream and the Philippine Sea anticyclonic anomalies in winter caused by El Niño connected with each other by local Hadley circulation over East Asia (Kang and Lee, 2017).

Previous studies have also found that the EAWM tended to be weak in El Niño winter (e.g., Chen et al., 2000; Huang et al., 2012; Wang and Chen, 2014; Kang and Lee, 2017). Wang et al. (2000) has explored how ENSO influenced its upstream climate

in East Asia, and found that the Pacific-East Asian teleconnection (PEAT) was the key bridge. PEAT is a vorticity wave pattern that starts from the central Pacific and extends poleward and westward to East Asia. In Figure 7c, PEAT could be recognized from the 850 hPa wind anomalies with a cyclonic vorticity over the central Pacific, the anticyclonic vorticities over western North Pacific, and a cyclonic vorticity over northeast Asia. The PEAT could be seen more clearly from the differences in the stream function at 500 hPa between El Niño and La Niña winters (Figure S4), which had an opposite sign with vorticity.

Corresponding to the anomalous anticyclones at 850 hPa over western North Pacific caused by El Niño, winter precipitation decreased over Indo-China Peninsula and northwestern Pacific, and increased over southern China (Figure 8a). In contrast, precipitation increased over Indo-China Peninsula and northwestern Pacific, and decreased over southern China during La Niña winter (Figure 8b). The opposite effects of El Niño and La Niña on the winter precipitation over southern China could be seen more clearly in Figure 8c, which was in accordance with the observational relationship between ENSO index and China precipitation in winter (http://cmdp.ncc-cma.net/pred/cn_enso.php?product=cn_enso_corr&season=DJF#corr). The decrease in winter precipitation over southern China caused by La Niña was very likely the reason for the more-than-normal haze days over the region during La Niña winter (Figure 4c), as less precipitation meant slower cleaning particles out of the atmosphere, which will be discussed in the next section (Figures 11b and 11c). Another reason probably could not be neglected that drier conditions over southern China during La Niña winter could avoid mistaking haze to fog days.

### 3.2.3 The effects of ENSO on winter atmospheric contents of aerosols

In this section, the changes in the winter aerosol CONCsur and loadings over China caused by ENSO were presented, and then the mechanism how ENSO affected the winter atmospheric contents of aerosols was analyzed from the perspective of wet and dry depositions and interregional transports of aerosols.

It was seen from Figure 9a that the winter CONCsur of aerosols were decreased by El Niño over northeastern China and eastern China. The winter aerosol CONCsur were also decreased by La Niña over northeastern China and eastern China, as well as the north part of northern China and most areas south of the Yangtze river (Figure 9b). Over southern China, although the increases in the winter CONCsur of aerosols caused by El Niño were not very clear in Figure 9a, they were obvious by comparing with La Niña winter (Figure 9c), which was generally in line with the simulation result of Feng et al. (2016). It was found from Figure 9d that the winter aerosol loadings were increased by El Niño over most areas east of 105°E and south of

40°N, and decreased over northeastern China and the north part of northern China. From Figure 9e, it was seen that the anomalous winter aerosol loadings caused by La Niña presented a meridional "- + -" pattern over the east part of China. The different influences of El Niño and La Niña on the winter aerosol loadings were the most obvious over the large areas south of the Yangtze river (Figure 9f).

From Figures 9c and 9f, it was found that the differences in the atmospheric contents of aerosols between El Niño and La Niña winters were more obvious over southern China than other areas of the country. To supply more information, Figures 9c and 9f were plotted in another way with different significance levels (Figure S5), which also showed that ENSO affected the winter atmospheric contents of aerosols over southern China more obviously than it did over other areas of the country. This was to some extent in line with the results in Figure 4 that only the winter haze days over southern China had significant relationship with ENSO. Therefore, in the following analysis, we focused on the mechanism how ENSO affected the winter atmospheric contents of aerosols over southern China.

It has been discussed in section 3.1 that the haze days over southern China tended to be less (more) than normal in El Niño (La Niña) winter, which was to some extent contradictory to the increase (decrease) in the winter atmospheric contents of aerosols over southern China caused by El Niño (La Niña). In the following of this section, we firstly tried to explain why the changes in the winter haze days and atmospheric contents of aerosols caused by ENSO over southern China were not consistent with each other. Then, we explored the reasons for the increase (decrease) in the winter atmospheric contents of aerosols caused by El Niño (La Niña) over southern China.

It has been revealed statistically in Figure 4c that southern China tended to suffer more (less) haze days than normal in La Niña (El Niño) winter, which was also seen in the 4 selected pairs of El Niño and La Niña winters (Figure S2) with one of them near 2010 (in numerical simulations, the aerosol emissions were fixed in 2010). However, numerical results showed that La Niña (El Niño) caused a decrease (increase) in the winter atmospheric contents of aerosols over southern China (Figure 9). Is it possible for southern China to have more (less) haze days but less (more) atmospheric contents of aerosols than normal in La Niña (El Niño) winter? To answer this question, the frequency distributions of the simulated winter daily aerosol CONCsur averaged over southern China (21°N–27°N, 104°–118°E, see Figure S1) in the groups of CLI, EL, and LA were plotted in Figure 10. It has been illustrated in section 3.2.1 that the simulated aerosol CONCsur in this study were smaller than

observational studies by 1~2 orders of magnitude. Therefore, for calibration, the simulated winter daily aerosol CONCsur over southern China were amplified by 10 times before plotting the frequency distributions.

It was seen from Figure 10 that the frequency distribution of the simulated winter daily aerosol CONCsur over southern China was a little right-skewed in the group of CLI, reaching peak at around 65 μg m$^{-3}$. The frequency distribution of the simulated winter daily aerosol CONCsur over southern China in the group of LA was a little left-shifting compared with that in CLI when aerosol CONCsur was larger than 40 μg m$^{-3}$. The frequency distribution of the simulated winter daily aerosol CONCsur over southern China in EL was a little right-shifting compared with that in CLI. The average numbers of days in winter with aerosol CONCsur bellow 40, between 40~80, and above 80 μg m$^{-3}$ in CLI, EL, and LA were also given in the top right corner of Figure 10. It was found that the number of days in winter with aerosol CONCsur between 40~80 μg m$^{-3}$ was the largest in LA and smallest in EL. Whereas, the number of days in winter with aerosol CONCsur above 80 μg m$^{-3}$ was the largest in EL and smallest in LA. Figure 10 indicated that southern China tended to have less clean and heavy but more moderate haze days than normal in La Niña winter. Whereas, in El Niño winter, southern China tended to have more heavy but less clean and moderate haze days than normal. This explained why southern China had less (more) haze days but more (less) atmospheric contents of aerosols than normal in El Niño (La Niña) winter. It should be noted that here we used regional mean aerosol CONCsur over southern China subjected to the low model spatial resolution, which was the reason why the curves in Figure 10 were close to each other. In the real practice, however, haze days and their severities are judged station by station (e.g., Figure 2), and the thresholds are not necessarily the same with what we used in Figure 10. Therefore, we can get some rough information about the effects of ENSO on the winter haze days over southern China from the shifts of curves in Figure 10, but cannot compared it with observational data directly.

As the emissions of aerosols were kept the same in all experiments (section 2.2), how quickly aerosols were removed from the atmosphere, especially through wet deposition, could affect the atmospheric contents of aerosols greatly (Zhang et al., 2016). It was found that ENSO influenced the winter wet depositions more obviously than the winter dry depositions of aerosols over China (Figure 11). The winter dry depositions of aerosols were decreased both by El Niño and La Niña over southern China (Figures 11c–f). Compared with dry deposition, wet deposition is a faster process. The winter wet depositions of aerosols over southern China were increased (decreased) by El Niño (La Niña) (Figures 11a–c), corresponding to the changes in winter

precipitation (Figure 8). Comparing Figure 9 and 11, it was found that the winter atmospheric contents and wet depositions of aerosols were both increased (decreased) by El Niño (La Niña) over southern China. It seemed that the changes in the winter wet depositions of aerosols were the results rather than the reasons of the changes in the winter atmospheric contents of aerosols over southern China.

Besides local emissions and removals, the interregional transports of aerosols could also influence the atmospheric contents of aerosols over a specific region. It was mentioned in Zhang et al. (2016) that South and Southeast Asia had become the important source areas of anthropogenic aerosols in 2010, which was also seen in the simulated aerosol CONCsur and loadings over these regions (Figure 6). A low-level anomalous anticyclone (cyclone) was caused by El Niño (La Niña) over the Philippine Sea in winter (Figure 7). The southwesterly (northeasterly) at the northwest of the anomalous anticyclone (cyclone)

led to an enhanced (weakened) transports of aerosols from South and Southeast Asia to southern China in El Niño (La Niña) winter (Figure 12). As the changes in the winter atmospheric contents of aerosols over southern China caused by ENSO could not be explained by local emissions or removals, it could only be attributed to the changes in the transports of aerosols from South and Southeast Asia to southern China. Zhu et al. (2012) has also pointed out that in determining aerosol concentrations, the changes in monsoon circulation were more dominant than that in precipitation (or wet deposition of particles) in East Asia.

It is expected that when the emissions of aerosols over South and Southeast Asia diminish in the future, the contradiction between the influences of ENSO on the winter haze days and atmospheric contents of aerosols over southern China will also disappear.

## 4 Conclusions and discussions

The effects of ENSO on the winter haze days and atmospheric contents of aerosols over China were discussed statistically and

numerically. Statistical results showed that southern China tended to have less (more) haze days than normal in El Niño (La Niña) winter, which was in line with the simulated more (less) winter precipitation over southern China caused by El Niño (La Niña). Statistical results indicated that the relationships between the winter haze days over northern and eastern China and ENSO were not significant. Numerical results also revealed that the influences of ENSO on the winter atmospheric contents of aerosols over northern and eastern China were not that obvious as over southern China. As a tropical phenomenon, it seemed

that ENSO affected the winter haze pollution over southern China more significantly than it did over northern and eastern China.

Numerical results indicated that the atmospheric contents of aerosols over southern China were more (less) than normal in El Niño (La Niña) winter, which was to some extent not in line with the effects of ENSO on the winter haze days of the region. In 2010, South and Southeast Asia have become the important source areas of anthropogenic aerosols (Zhang et al., 2016). The enhanced southwesterly (northeasterly) at the northwest of the winter anomalous anticyclone (cyclone) over the Philippine Sea caused by El Niño (La Niña) enhanced (weakened) the transports of aerosols from South and Southeast Asia to southern China. The frequency distributions of the simulated daily surface concentrations of aerosols in winter over southern China indicated that the region tended to have more heavy (moderate) haze days, but less clean and moderate (heavy) haze days than normal in El Niño (La Niña) winter. But it should be noted again that the emission data of aerosols used in our study were fixed in 2010, and if the emissions of aerosols change the story may be different.

Many studies have found that the haze pollution over northern and eastern China was influenced by the EAWM, as we have cited in section 1. And previous studies also found that ENSO could influence the strength of the EAWM, as we have discussed in section 3.2.2 that El Niño could weaken the EAWM. It was expected that there would be some connection between ENSO and the winter haze pollution over northern and eastern China. However, the connection between ENSO and the winter haze pollution over northern and eastern China was not strong, e.g., the correlations between the winter haze days of the two regions and ENSO were not significant. This made us to think the way how haze pollutants over the north part of China were cleared away in winter. Usually, a few clean days in the north part of China followed a breakout of a block over high latitude Asia, during which a stream of cold air swept southwardly and rapidly from the Siberia or Mongolia. Therefore, we could get some information about the influence of ENSO on the winter haze pollution over the north part of China from the frequency of cold air recorded in China in El Niño and La Niña winters. In the Decembers of 1986 (during the 1986/1987 El Niño event) and 2010 (during the 2010/2011 La Niña event), cold air happened in China both for 7 times, the maximum of the same month during 1960–2011 (Figure S6). This indicated that El Niño (La Niña) did not necessarily reduce (increase) the cold air frequency of China in winter, even though it could weaken (strengthen) the EAWM. To the contrary, it seemed that El Niño (La Niña) was in favor of (not in favor of) the cold air frequency in China in winter (Figure S7). The relationship between

ENSO and the cold air frequency of China in winter made the connection between ENSO and the winter haze pollution over the north part of China more complicated than it was expected simply from the perspective of the EAWM. The cold air frequency of China in winter is also influenced by the upstream conditions in the Atlantic Ocean, the Arctic ice, and the Eurasian snow cover (not shown). This emphasizes the importance of exploring the comprehensive effects of ENSO and the extratropical systems on the haze pollution of China.

Haze pollution is a very sophisticated problem, because it is the comprehensive result of human activities and weather conditions. Weather conditions determine the dispersions and removals of haze particles over a specific region, influence the complex chemistry reactions among different components, and also connect the haze pollution of a specific region with the emissions of neighboring regions. To be more complicated, haze pollution and weather conditions interact with each other closely over some areas. This work explores the effects of ENSO on the winter haze pollution over China under relatively simple experimental settings, with prescribed SST and fixed aerosol emissions, which means that SST does not response to aerosols. Besides, the chemistry reactions in the model we used is also simplified, especially without the complex reactions related to nitrate aerosols. Therefore, studies with more sophisticated experimental designs (e.g., with atmosphere and ocean coupled models) and chemistry schemes are still in need in the future as to the topic discussed in this work.

**Author contributions.** The work was done under the guidance of Hua Zhang. Shuyun Zhao and Hua Zhang designed the experiment, and prepared the manuscript. Bing Xie conducted model simulations. The analysis of results and preparation of all figures and tables were done by Shuyun Zhao.

**Acknowledgement**

This work was financially supported by the (Key) National Natural Science Foundation of China (91644211&41575002). And we would like to thank Dr. Shao Sun of the National Climate Center, Chinese Meteorological Administration, for his kind help in plotting figures.

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

**Tables and Figures**

Table 1. Simulation set-up.

| Group name | SST | Running time | Output frequency |
|:---:|:---:|:---:|:---:|
| **CLI** | Climatologic SST | Oct.[0]−Aug.[1] | Monthly & Daily |
| **EL** | Climatologic SST + $\Delta SST_{El\ Niño}$ | Oct.[0]−Feb.[1] | Monthly & Daily |
| **LA** | Climatologic SST + $\Delta SST_{La\ Niña}$ | Oct.[0]−Feb.[1] | Monthly & Daily |

The superscripts of the 3rd column: 0 and 1 represent the 1st and 2nd model year, respectively.

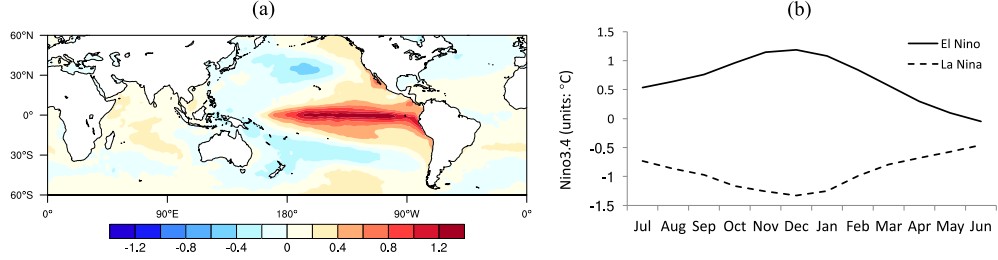

**Figure 1. (a) Typical ENSO mode (units: °C/°C) and (b) average monthly Niño3.4 (units: °C) of 21 El Niño and 18 La Niña events from 1951 to 2015.**

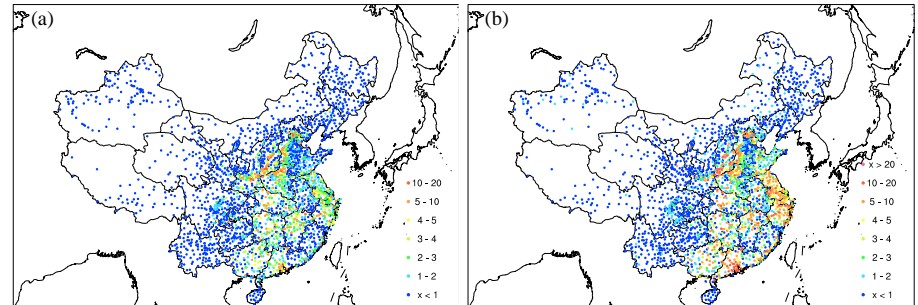

**Figure 2. The winter-average monthly haze days (units: days/month) during the years of (a) 1960-2013 and (b) 2000-2013 over main land China.**

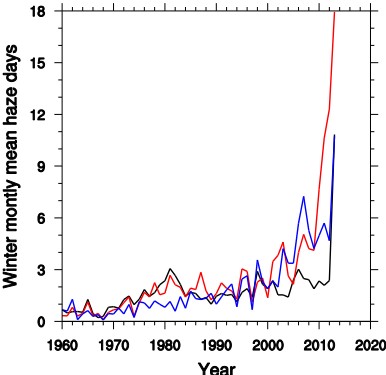

**Figure 3. Time series of the winter-average monthly haze days (units: days/month) of JJJ (black), JZH (red), and GG (blue).**

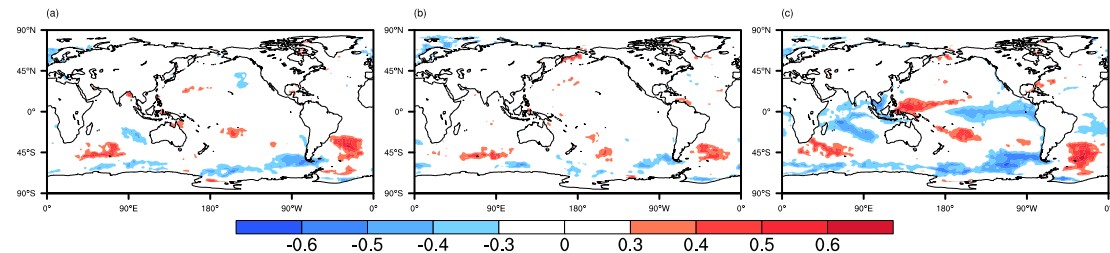

**Figure 4. Correlation coefficients (unitless) of the monthly haze days of (a) JJJ, (b) JZH, and (c) GG with SST in winter, after applying a band-pass filtering of 2-8 years to both the data of haze days and SST. Shade denotes that results pass 95% significance level.**

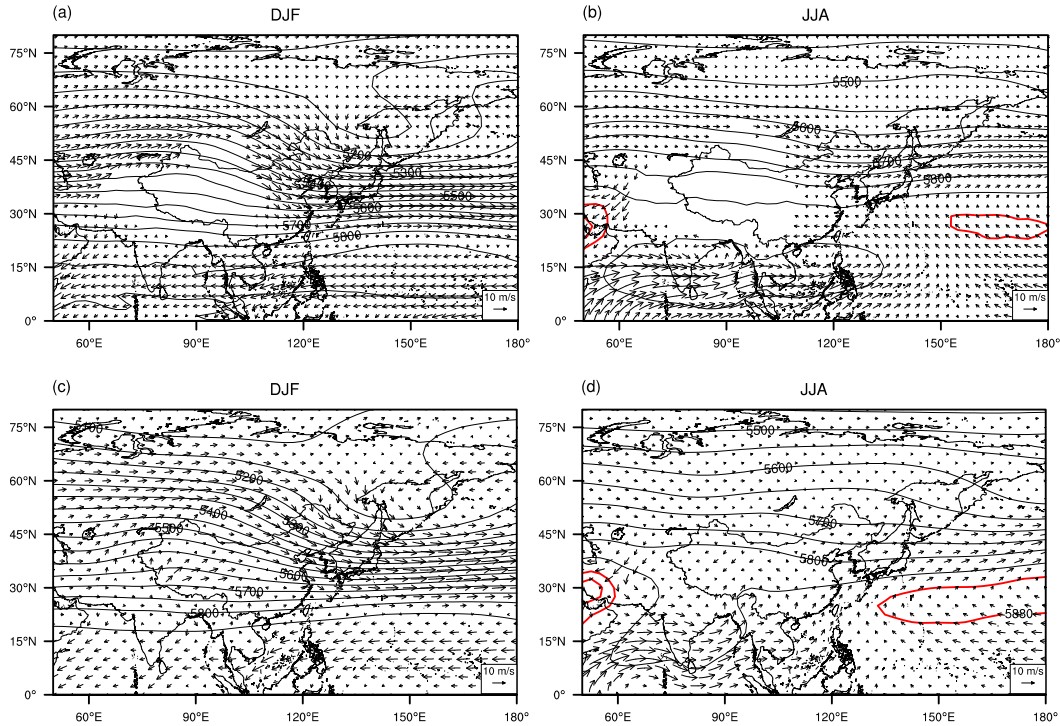

**Figure 5.** Comparisons between (a–b) the simulated and (c–d) reanalysis winter and summer geopotential height at 500 hPa (contour, units: gpm) and wind at 850 hPa (vector, units: m s$^{-1}$), with the blank places in (a) and (b) are because of the influence of the Tibetan Plateau. Model results are from the group of CLI, and reanalysis data are from NCEP.

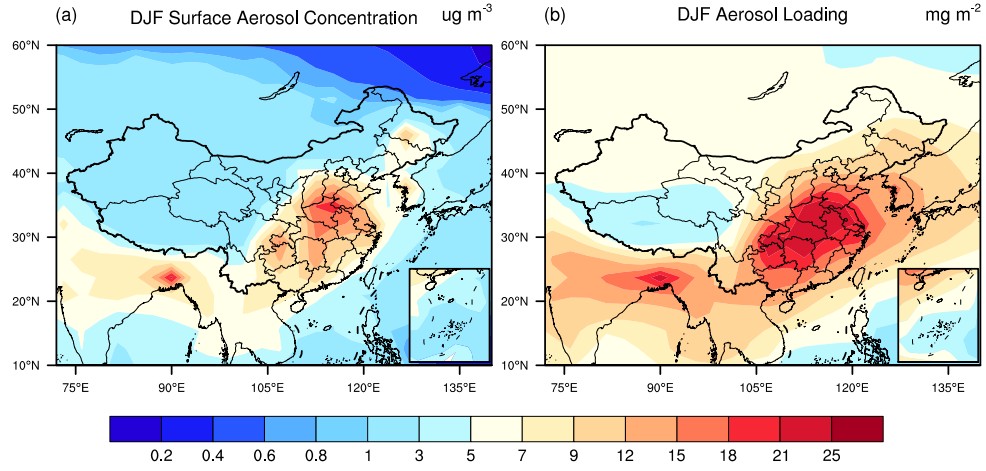

**Figure 6.** The simulated winter-average (a) surface aerosol concentrations (units: μg m$^{-3}$) and (b) aerosol loadings (units: mg m$^{-2}$) in

**the group of CLI.**

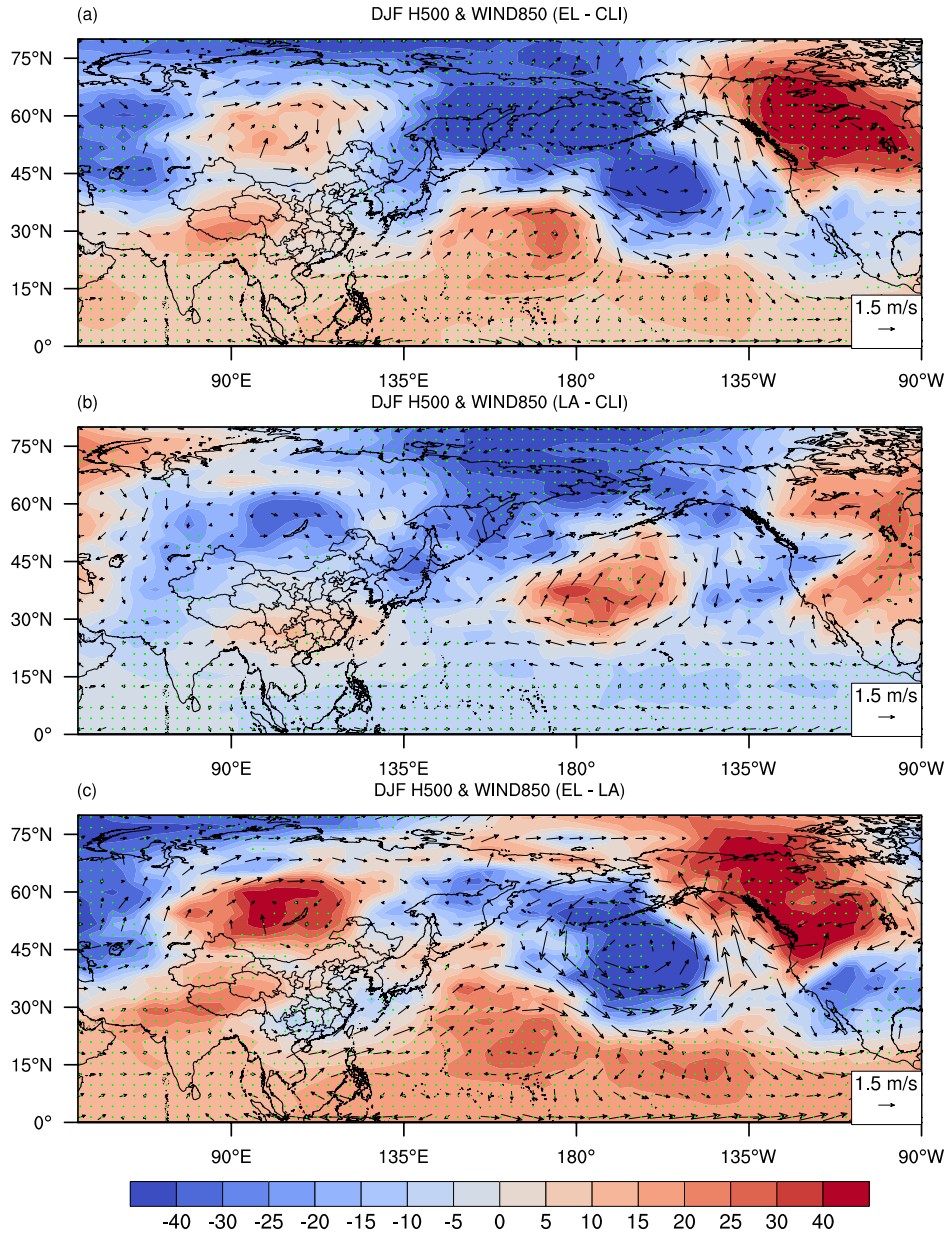

**Figure 7. Medians of the simulated differences in winter-average geopotential height at 500 hPa (H500, shaded, units: gpm) and wind at 850 hPa (WIND850, vector, units: m s⁻¹) between (a) EL and CLI, (b) LA and CLI, and (c) EL and LA, with green dots indicating that the differences of H500 between more than 70% pairs of ensemble members have the same sign with the median**

**differences.**

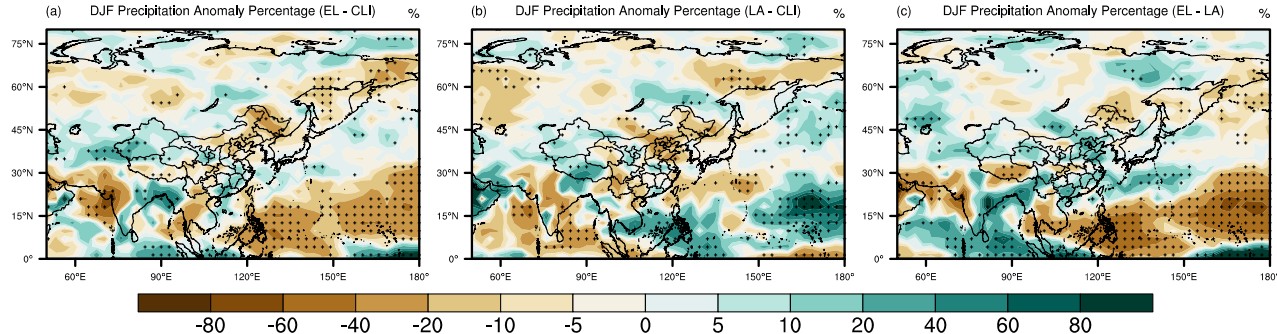

**Figure 8. Medians of the simulated differences in winter-average precipitation in percentage (units: %) between (a) EL and CLI, (b) LA and CLI, and (c) EL and LA, with black dots indicating that the differences between more than 70% pairs of ensemble members have the same sign with the median differences.**

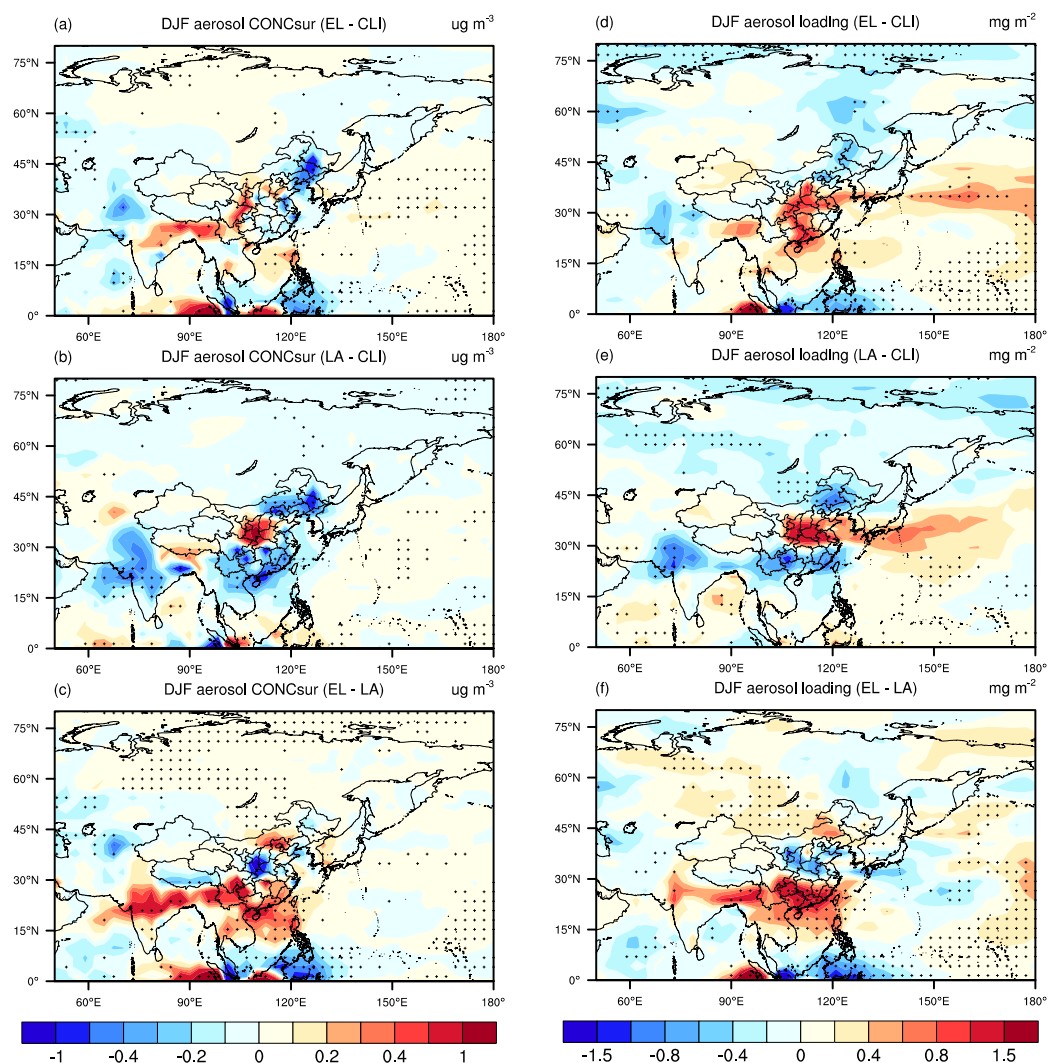

**Figure 9.** Medians of the simulated differences in winter-average aerosol surface concentrations (left panel, units: μg m⁻³) and aerosol loadings (right panel, units: mg m⁻²) between (a)~(d) EL and CLI, (b)~(e) LA and CLI, and (c)~(f) EL and LA, with black dots indicating that the differences between more than 70% pairs of ensemble members have the same sign with the median differences.

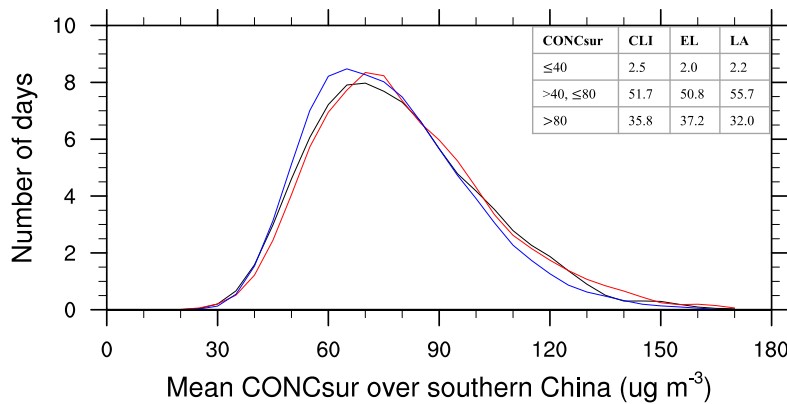

| CONCsur | CLI | EL | LA |
|---|---|---|---|
| ≤40 | 2.5 | 2.0 | 2.2 |
| >40, ≤80 | 51.7 | 50.8 | 55.7 |
| >80 | 35.8 | 37.2 | 32.0 |

**Figure 10. Frequency distributions of the simulated winter daily surface aerosol concentrations (CONCsur, units: μg m⁻³) averaged over southern China (after being amplified by 10 times), with black, red and blue lines representing the results from CLI, EL, and LA, respectively; The average numbers of days in winter with CONCsur ≤ 40, 40 <CONCsur ≤ 80, and CONCsur>80 in CLI, EL, and LA are given in the table at the top right corner.**

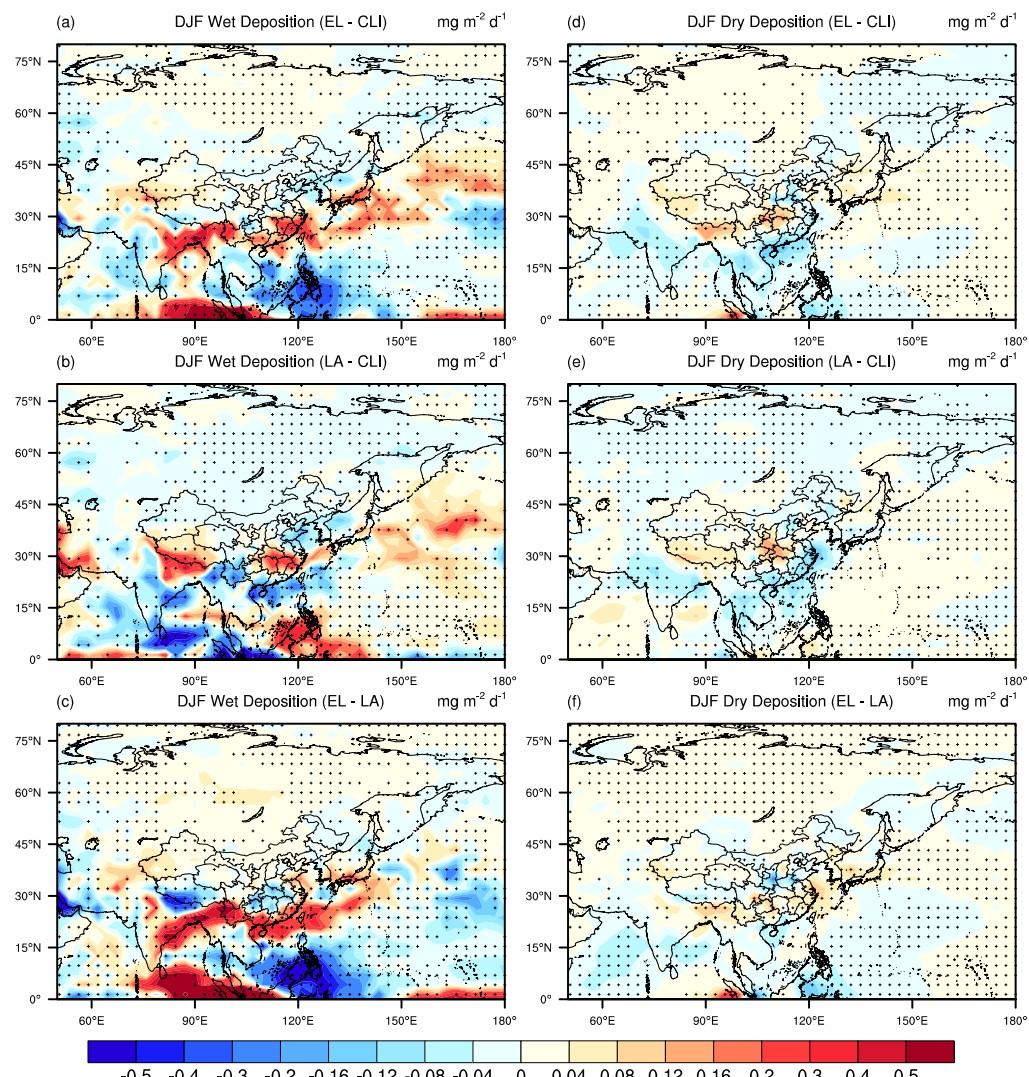

**Figure 11. Medians of the simulated differences in winter-average wet depositions (left panel) and dry depositions (right panel) of aerosols (units: mg m⁻² d⁻¹) between (a)~(d) EL and CLI, (b)~(e) LA and CLI, and (c)~(f) EL and LA, with black dots indicating that the differences between more than 70% pairs of ensemble members have the same sign with the median differences.**

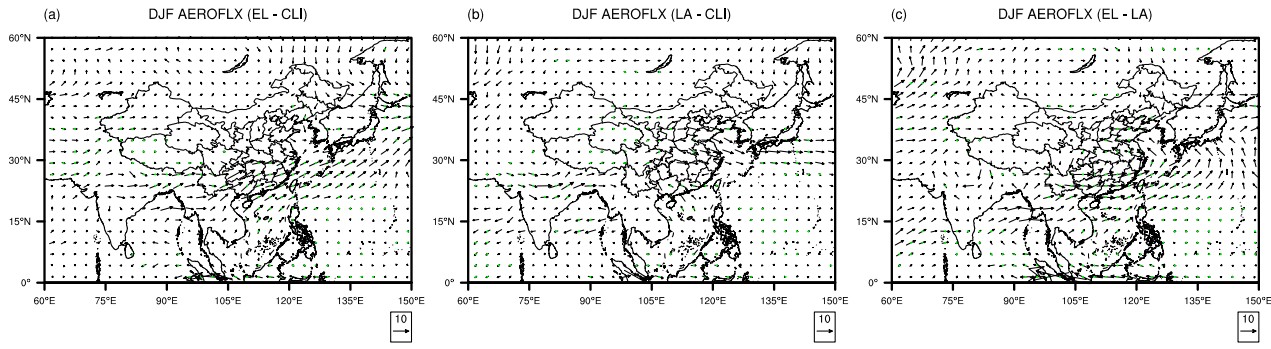

**Figure 12. Medians of the simulated differences in the winter-average vertical integral of aerosol horizontal fluxes (AEROFLX, units: kg m⁻¹ s⁻¹) between (a) EL and CLI, (b) LA and CLI, and (c) EL and LA, with green dots indicating that the differences of the zonal components of AEROFLX between more than 70% pairs of ensemble members have the same sign with the median differences.**

