# Peer review of "The Effects of El Niño-South Oscillation on the Winter Haze"

_Atmospheric Chemistry and Physics, 2017_

## Referee Comment (RC1) · Anonymous Referee #1 · 24 Jul 2017

The manuscript explores the potential for the ENSO to have influenced the winter haze days and contents of anthropogenic aerosols in China. The observational data and the numerical simulations that was designed to isolate the impacts of ENSO point to an contrasting effect between El Nino and La Nina conditions on changes in the haze days and aerosol concentration over the southern China. However, the analysis and presentation shown in this paper, I think, have several flaws. Therefore, the revision is required to improve the reliability of the results reported.

General comments

1.Statistical significance: The significance of results has not been discussed except for Fig.4. I recommend the authors show the significance in the figures. Because the authors have conducted multiple ensemble experiments, I think it is possible to do so.

[Figure]

2.Use of emission data for the year of 2010: The aerosol emissions had a long-term trend, the decadal variability, and regional deference. So, I would encourage the authors to check and argue the results for the other years.

3.My understanding is that the effects of ENSO on the haze pollution (or haze seasonal prediction) in southern China depend on the changes in aerosol transport and precipitation associated with changes in the SST patterns. However, I am very confused about the contrasting features of "haze days" and "aerosol concentrations" (Fig.4 and Fig. 9, P1. L19-20, P12. L16-18). I have several questions. Which is a larger damage by haze pollution in El Nino years (less haze days but heavy) or La Nina years (many haze days but moderate) eventually? If it were less precipitation in El Nino years, one can assume that winter haze days in El Nino years increase?

Specific comments

1. P3.L13: According to the CAM definition of the haze, relative humidity (<80%) is included in it. Dose this definition implies that the haze days are hard to be counted in the El Nino years because of the rainy years?

2. Figs. 2,3: I ask the authors to show separately the haze days during the El Nino or La Nina years.

3. Representation of geographical names in China (Fig.2, P5): The paper makes use of many area names in China. However, since general readers don't know their location, I would request the authors to plot the important locations in Fig.2.

4. P5. L17-19: The three regions (JJJ, JGZ, GG) are hard to grasp for me. These areas should be shown in the Figures.

5. Fig. 5: I don't understand why the authors represent the mean states in JJA. I think the analysis of the JJA is unnecessary because the paper focuses on the winter season.

6. P10. L7, Fig.10: The authors make a point that "the winter-average wet depositions

of aerosol over southern China are decreased by La Nina (Fig.10b)". However the wet depositions seem to be increased over the southern China during La Nina, although they are decreased over the south coast of China and the sea.

7. Fig.12: I suggest that the each number of days categorized by "moderate" or "heavy" aerosol concentrations are shown during El Nino and La Nina in addition to the PDF, because I have difficulty reading the small difference between the ENSO from the PDF.

8. P1. L20 and P12. L16-17: Do you use the "aerosol " as a synonymous term for "haze" ? It is confusing for me because the authors do not mention the relationship between aerosol concentration and occurrence potential of haze days.

---

## Short Comment (SC1) · 26 Jul 2017

In this manuscript, the authors found observational evidence about the regional dependency of the relationship between winter haze days of China and ENSO SST, and tried to verify the mechanism about the effect of ENSO on the winter haze pollution of China based on the composite analysis for El Nino cases and La Nina cases which are simulated by using the aerosol-climate coupled model BCC_AGCM2.0_CUACE/Aero of the NCC, focusing both transport effect and scavenging effect which are related to the atmospheric circulation change and associated precipitation change induced by different SST forcing. The methodology is sound and the paper appears to be relatively well-organized. I believe that the paper provides a further insight into the regional dependency of the ENSO effect on haze pollution of China. However in this version

there are some scientific comments to be clarified.

Specific comments

1. Significant test: the most important procedure in composite analysis is to provide readers with the significant test for the difference in target variables between two groups, at least 5% significant level. So, the authors should add the significant level for every difference field. The simulated changes caused by El Nino and La Nina forcing relative to climatology are not always symmetric each other. So, it is very difficult to understand the difference between El Nino effect and La Nina effect. In my opinion the other approach to stress the effect of El Nino on the winter haze is to see the difference between two extremes (El Nino and La Nino). In this way, there will be a clear difference by offsetting the same sign and magnitude. Anyway the authors should revise the manuscript based on the significant test. In addition the authors should add some discussions about the dynamic link between circulation change and ENSO forcing through telelconnection pathway based on the significant test.

2. Filtering of the monthly haze days: In Fig.4 the authors calculated the correlation coefficients between the winter-average monthly haze days and SST in winter, using data after applying a linear-trend removing and 2~8 years' band-pass filtering. Basically 2-8 years' band-pass filter also eliminates a linear trend because periods less than 2 years and over 8 years including a linear trend are removed. So, using both filters is redundant. Applying a linear-trend removing at first is obviously not valid because time series is not linear. Moreover sudden jump in last year can make linear trend distorted toward large trend. Thus the author should clarify this point. If it is so, the authors should re-calculate the correlation coefficients in Fig. 4 by considering the point suggested above.

3. Haze days and surface aerosol concentration in Southern China: Is less haze days in observation during El Nino winter shown in Fig. 3c matched with less surface aerosol concentration in model during El Nino winter shown in Fig. 9a? If it is true, why are

surface aerosol concentrations over southern China decreased in both El Nino and La Nina winter? Please clarify this point. In addition, the authors should check whether Fig 9a and b are consistent with Fig. 12 over southern China. In my opinion, two pdf curves for El Nino and La Nina should be located below CLI curve on average because changes in both Fig. 9a and b are negative over southern China. Are pdf results dependent on the definition of southern China?

Technical corrections

1. Line 9 page 8: "the weakness of EAWM" should be "the intensification of EAWM" because Siberian high is stronger than normal.

2. Line 5-8 page 9: Over northeastern China winter CONCsur of aerosols are decreased in both El Nino and La Nina, indicating that the decrease over northeastern China is not irrelevant to ENSO. So the authors should describe this point carefully.

3. Line 1 page 24: "zonal flux" should be "horizontal flux".

---

## Referee Comment (RC3) · Anonymous Referee #3 · 30 Jul 2017

Using observations and model simulations, the authors investigated the possible influences of ENSO events on aerosols over China. Understanding the changes in aerosols is a relevant topic for improving our knowledge of relationship between natural cycle and aerosols. Based on the observational data, they concluded that the haze days of southern China tend to be less (more) than normal in El Niño (La Niña) winter, however the relationship is not significant. However, inconsistent result is seen based on the simulated result. And they concluded it is due to the fact that heavy haze days are more frequent in El Nino winter. I agree with the authors that the influence of ENSO events on aerosol concentrations over China cannot be ignored. And it is an important interdisciplinary issue which needs more attention and deep researches. However, there are many problems of this manuscript, and it cannot be accepted by ACP as it is

now. Some specific comments or suggestions are listed as follows.

1. The authors mainly focused on the influence of ENSO via EAWM on the haze over China, however, what is exact influences of the ENSO influence on the EAWM? And the involved physical processes. The authors did not give a detailed explanation regarding this point neither concluded the existed explanations. I suggest the authors do not avoid discussing this question, but make complete statement about this point. 2. The influences of ENSO on the aerosols have been discussed in recent works. Regarding the impacts of El Niño and La Nina Modoki events on aerosol concentrations over eastern China, and the impacts of EASM on O3 over eastern China. The authors should include the relevant work into the present study, compare the differences, and further highlight the motivation of the present study. 3. The abbreviations of JJJ, JZH, and GG show no sense, the abbreviations should include geographic information and easy remember. I suggest using north China, southern China to replace. 4. The contradict description in the manuscript, for example, P5L21, "After the year of 2000, the winter-average monthly haze days over the three regions grew dramatically, especially over JZH and GG"; but in P5L25, "haze days over JZH and GG increased too abruptly after the year of 2010, especially in 2013". 5. The reasons for the applying a linear-trend removing and a 2-8 years band-pass filtering showed be illustrated, and the valid freedom of significant should be applied to the significance test. In Figure 3, it is seen that the hazes days show nonlinear variations, even within 1950-2010. And whether the hazes days show a 2-8 years periods is not clear, it makes me hard to understand why the authors removed the linear trend and a band pass filter. 6. As to the influences of ENSO on the EAWM, many researchers have illustrated that the ENSO shows important role in impacting the intensity of EAWM, and the climate, including temperature, rainfall and winds over north China, central China are impacted by the EAWM. However, the result shows the role of ENSO on hazes over JJJ and JZH is not evident. I suggest the authors to separate the stronger and moderate ENSO events, or separate the eastern and central ENSO events to further explore the result, and it is unknown the exact years of El Nino and La Nina in the present work. 7. The performance of

the model used to perform the simulations in reproducing ENSO patterns as well as its variations is unknown. Since ENSO is a complex air-sea interaction process, it is necessary to access the performance in simulating ENSO whereas given the circulations over eastern China during winter and summer. And the observed variations in rainfall should be shown while the simulated result for the poor simulations of rainfall in the models. 8. Figure captions, the blank in figure 5a, b, should be mentioned, however not in c and d. Figure 7, shaded for hgt not contour. Figure 9, it is better if the relative changes is shown whereas the absolute values, considering that the climatology mean shows big differences as shown in Figures 1 and 6.

---

## Author Comment (AC1) · 12 Oct 2017

Please see the supplementary ".zip".

Referee's comments and responses to them are in "reply2RC1.pdf".

Changes in the manuscript can be seen in "manuscript-with-changes.docx" and "supplementary-document.pdf".

We greatly appreciate these comments and suggestions.

Please also note the supplement to this comment:
https://www.atmos-chem-phys-discuss.net/acp-2017-506/acp-2017-506-AC1-supplement.zip

---

## Author Comment (AC2) · 12 Oct 2017

Please see the supplementary ".zip" file.

Referee's comments and responses to them are in the "reply2RC2.pdf".

Changes in the manuscript can be seen in the "manuscript-with-changes.docx" and "supplementary-document.pdf"

We greatly appreciate these comments and suggestions.

Please also note the supplement to this comment:
https://www.atmos-chem-phys-discuss.net/acp-2017-506/acp-2017-506-AC2-supplement.zip

---

## Author Comment (AC3) · 12 Oct 2017

It is found that the RC2 (Referee comment 2) is the same with SC1 (short comment 1). Therefore, author's reply to RC2 can be seen in the reply to SC1.

We greatly appreciate these comments and suggestions.

---

## Author Comment (AC4) · 12 Oct 2017

Please see the supplementary ".zip" file.

Referee's comments and responses to them are in the "reply2RC3.pdf".

Changes to the manuscript can be seen in the "manuscript-with-changes.docx" and "supplementary-document.pdf".

We greatly appreciate these comments and suggestions.

Please also note the supplement to this comment:
https://www.atmos-chem-phys-discuss.net/acp-2017-506/acp-2017-506-AC4-supplement.zip

---

## Author Response (AR1)

et al.

Anonymous Referee #1

**The manuscript explores the potential for the ENSO to have influenced the winter haze days and contents of anthropogenic aerosols in China. The observational data and the numerical simulations that was designed to isolate the impacts of ENSO point to a contrasting effect between El Nino and La Nina conditions on changes in the haze days and aerosol concentration over the southern China. However, the analysis and presentation shown in this paper, I think, have several flaws. Therefore, the revision is required to improve the reliability of the results reported.**

General comments

1.Statistical significance: The significance of results has not been discussed except for Fig.4. I recommend the authors show the significance in the figures. Because the authors have conducted multiple ensemble experiments, I think it is possible to do so.

Reply: We used the median instead of the ensemble mean of the differences of variables caused by El Niño or La Niña, as median is more robust and resistant to extreme values. And we also added markers to those grids where the differences of a specific variable between more than 70% pairs of ensemble members have the same sign with the median differences, as a reflection of significance. Please see the last paragraph of section 2.2.

2.Use of emission data for the year of 2010: The aerosol emissions had a long-term trend, the decadal variability, and regional deference. So, I would encourage the authors to check and argue the results for the other years.

Reply: Thank you for the suggestion. Actually, the effects of ENSO on the haze pollution of China under different emissions (even unrealistic large emissions) are parts of what we are considering to do in our next work.

3.My understanding is that the effects of ENSO on the haze pollution (or haze seasonal prediction) in southern China depend on the changes in aerosol transport and precipitation associated with changes in the SST patterns. However, I am very confused about the contrasting features of "haze days" and "aerosol concentrations" (Fig.4 and Fig. 9, P1. L19-20, P12. L16-18). I have several questions. Which is a larger damage by haze pollution in El Nino years (less haze days but heavy) or La Nina years (many haze days but moderate) eventually? If it were less precipitation in El Nino years, one can assume that winter haze days in El Nino years increase?

Reply: It is very hard to tell which is more damaging, less haze days but some of them are very heavy or more haze days but they are generally moderate. This manuscript only discussed a phenomenon. The social and economic effects of haze pollution is a little beyond our research field. Personally, I think we cannot count on a natural circulation to release our suffering from haze pollution, instead we should reduce the emissions of aerosols and their precursors.

Precipitation enhances the wet deposition of particles, which is an important way of cleaning particles out of the atmosphere. In southern China, it was found that there were more precipitation and less haze days in El Niño winters. The changes in precipitation and haze days correspond with each other very well. But this correspondence may not be good over other places. For example, in northern China, there are always a few clean days after an outbreak of a block in the high latitude, even without rain in winters, because strong northerly wind blows the haze pollutants away from northern China. That is why many studies related the winter haze events of northern and eastern China to Arctic sea ice and Eurasian snow cover. So, we can say that precipitation facilitates the reduction of haze days, but at the same time we must also check other meteorological variables for a specific region.

Specific comments

1. P3. L13: According to the CAM definition of the haze, relative humidity (<80%) is included in it. Dose this definition implies that the haze days are hard to be counted in the El Nino years because of the rainy years?

Reply: For a long time, meteorological stations use relative humidity to distinguish haze and fog when visibility is low (< 10 km), before the large-scale installation of precise instrument that can detect particle matters. The reason is that when humidity

is very high, it is easy for particles to activate as the cores of fog droplets. We do agree that there might be possibilities to mistake haze to fog, or vice versa, especially when the relative humidity is around the threshold (80%). That's why we mentioned in the manuscript that "Another reason probably cannot be neglected that drier conditions over southern China during La Niña winter can avoid mistaking haze to fog days". However, though maybe with some errors, the CMA haze-day dataset is still an authorized data set with long time period and large spatial coverage.

2. Figs. 2,3: I ask the authors to show separately the haze days during the El Nino or La Nina years.

Reply: In the last more than 50 years, the emissions of haze particles and their precursors have experienced dramatic variations, which can blanket the differences between the haze days in El Nino and La Nina winters. So, we selected 4 pairs of El Nino and La Nina winters, with the time interval of each pair ≤ 2 years, in order to diminish the influence of emission variations as much as possible. Please see the revision and the Figure S2 in the supplementary material.

3. Representation of geographical names in China (Fig.2, P5): The paper makes use of many area names in China. However, since general readers don't know their location, I would request the authors to plot the important locations in Fig.2.

Reply: Fig.2 is too crowed for other marks, so we marked the locations of the provinces mentioned in section 3.1 in Figure S1 in the supplementary material.

4. P5. L17-19: The three regions (JJJ, JGZ, GG) are hard to grasp for me. These areas should be shown in the Figures.

Reply: We also marked the areas of the three regions in Figure S1 in the supplementary material.

5. Fig. 5: I don't understand why the authors represent the mean states in JJA. I think the analysis of the JJA is unnecessary because the paper focuses on the winter season.

Reply: Monsoon circulation is the key factor to understand the influences of ENSO over East Asia. Therefore, the first thing we thought about was to explore if the model could capture the general characteristics of the East Asian monsoon circulations. We gave the monsoon circulations in winter as well as in summer in order to comprehensively explore this question. The

common problem in simulating the circulations of the East Asian winter monsoon and East Asian summer monsoon by the model might give some clues in the future works, e.g., improving the model or using the model in other studies.

6. P10. L7, Fig.10: The authors make a point that "the winter-average wet depositions of aerosol over southern China are decreased by La Nina (Fig.10b)". However, the wet depositions seem to be increased over the southern China during La Nina, although they are decreased over the south coast of China and the sea.

Reply: In many studies, people used "southern China" to represent the large area of China south of the Yangtze river. But in the climate category of the National Climate Center of China Meteorological Administration (NCC/CMA), southern China means the region that generally includes the provinces of Guangdong and Guangxi, very similar to the region of GG in our work (Figure S1).

7. Fig.12: I suggest that each number of days categorized by "moderate" or "heavy" aerosol concentrations are shown during El Nino and La Nina in addition to the PDF, because I have difficulty reading the small difference between the ENSO from the PDF.

Reply: We have changed the PDF to frequency distribution, so that the absolute number of days can be seen from the figure directly. And we also gave the number of days with the regional mean surface aerosol concentration over southern China (CONCsur) $\leq$40, 40<CONCsur$\leq$80, and CONCsur>80 $\mu g/m^3$. Please see the revised Figure 12. But it is necessary to note that here we used regional mean CONCsur subjected to the low spatial resolution of the model (2.8º x 2.8º). In the real operation, haze days and their degrees of severity are judged station by station. Therefore, Figure 12 is given just to explain a possibility that southern China can have less (more) haze days but more (less) mean aerosol concentration in El Nino (La Nina) winters, through analyzing the shifts of the curves in El and LA relative to the CLI.

8. P1. L20 and P12. L16-17: Do you use the "aerosol" as a synonymous term for "haze"? It is confusing for me because the authors do not mention the relationship between aerosol concentration and occurrence potential of haze days.

Reply: Haze pollution happens under stagnant weather conditions, because human-beings are emitting large amount of aerosol

particles and their precursors into the atmosphere. Therefore, in the part of model simulation, we used the aerosols that were mainly emitted by human-beings as a proxy of haze pollution. Comparing Figure 2 and 6, it can be seen that the distributions of the winter haze days and aerosol concentrations over China are consistent with each other. But, as we have discussed in the last paragraph of the manuscript that haze pollution is a very complicate problem, and works with more complicate experimental designs and chemistry schemes are still in need in the future.

**reviewer 2**
**In this manuscript, the authors found observational evidence about the regional de- pendency of the relationship between winter haze days of China and ENSO SST, and tried to verify the mechanism about the effect of ENSO on the winter haze pollution of China based on the composite analysis for El Nino cases and La Nina cases which are simulated by using the aerosol-climate coupled model BCC_AGCM2.0_CUACE/Aero of the NCC, focusing both transport effect and scavenging effect which are related to the atmospheric circulation change and associated precipitation change induced by different SST forcing. The methodology is sound and the paper appears to be relatively well-organized. I believe that the paper provides a further insight into the regional dependency of the ENSO effect on haze pollution of China. However, in this version there are some scientific comments to be clarified.**

Specific comments

1. Significant test: the most important procedure in composite analysis is to pro- vide readers with the significant test for the difference in target variables between two groups, at least 5% significant level. So, the authors should add the significant level for every difference field. The simulated changes caused by El Nino and La Nina forcing relative to climatology are not always symmetric each other. So, it is very difficult to understand the difference between El Nino effect and La Nina effect. In my

opinion the other approach to stress the effect of El Nino on the winter haze is to see the difference between two extremes (El Nino and La Nino). In this way, there will be a clear difference by offsetting the same sign and magnitude. Anyway, the authors should revise the manuscript based on the significant test. In addition, the authors should add some discussions about the dynamic link between circulation change and ENSO forcing through teleconnection pathway based on the significant test.

Reply: About the significance. In the part of model results, we used the medians instead of the ensemble means of the differences of variables caused by ENSO, as median is more robust and resistant to extreme value that might happen in a specific experiment. And we also added markers to those grids where the differences of a specific variable between more than 70% pairs of ensemble members have the same sign with the median differences, as a reflection of significance. Please see the revision.

According to the suggestion of the reviewer, we added discussions about the differences of variables between El Niño and La Niña winters. Please see the revision. Thanks for the suggestion.

About the teleconnection. We added discussions on how ENSO forcing influences the EAWM circulation through the Pacific-East Asian teleconnection, please see section 3.2.2 in the revised manuscript and the Figure S4 in the supplementary document.

2. Filtering of the monthly haze days: In Fig.4 the authors calculated the correlation coefficients between the winter-average monthly haze days and SST in winter, using data after applying a linear-trend removing and 2~8 years' band-pass filtering. Basically 2-8 years' band-pass filter also eliminates a linear trend because periods less than 2 years and over 8 years including a linear trend are removed. So, using both filters are redundant. Applying a linear-trend removing at first is obviously not valid because time series is not linear. Moreover, sudden jump in last year can make linear trend distorted toward large trend. Thus, the author should clarify this point. If it is so, the authors should re-calculate the correlation coefficients in Fig. 4 by considering the point suggested above.

Reply: Indeed, a 2-8 years' band-pass filtering has already eliminated a linear trend. Therefore, we replotted Fig. 4 only applying the band-pass filtering, please see the revision. Considering the abrupt increase of haze days after 2010, especially in 2013, we only used the data between 1960-2010. Please see the 4[th] paragraph of section 3.1.

3. Haze days and surface aerosol concentration in Southern China: Is less haze days in observation during El Nino winter shown in Fig. 3c matched with less surface aerosol concentration in model during El Nino winter shown in Fig. 9a? If it is true, why are surface aerosol concentrations over southern China decreased in both El Nino and La Nina winter? Please clarify this point. In addition, the authors should check whether Fig 9a and b are consistent with Fig. 12 over southern China. In my opinion, two pdf curves for El Nino and La Nina should be located below CLI curve on average because changes in both Fig. 9a and b are negative over southern China. Are pdf results dependent on the definition of southern China?

Reply: The increase in winter surface aerosol concentration (CONCsur) over southern China caused by El Niño is not very obvious in Fig. 9a. But if we compare the results between El Niño and La Niña winters, which we have added according to the suggestion of the reviewer, El Niño causes more aerosol CONCsur over southern China (revised Fig. 9c). And the increase in the winter atmospheric contents of aerosols over southern China caused by El Niño can be seen more clearly from the variable of aerosol loadings (revised Fig. 9, right panel). In this work, southern China mainly includes Guangdong and Guangxi provinces (the "GG" in Figure 3 and 4), according to the climate category of the National Climate Center of China Meteorological Administration. The scope for calculating regional mean aerosol CONCsur over southern China in Fig. 12 is given in the added supplementary document (Fig. S1), consistent with the southern China in Fig. 9.

According to the suggestion of another reviewer, we have changed the PDF in Fig. 12 to frequency distribution. As we have explained in the last paragraph but two of section 3 that here we used regional mean aerosol CONCsur, which caused the curves in Fig. 12 to look close to each other. However, we can still find that the curve for La Niña (El Niño) generally shifts leftward (rightward) comparing with that for CLI. And it can also be reflected from the top right corner in Fig. 12 that southern China tends to have more moderate (heavy) but less clean and heavy (moderate) haze days in La Niña (El Niño) winter.

Technical corrections

1. Line 9 page 8: "the weakness of EAWM" should be "the intensification of EAWM" because Siberian high is stronger than normal.

Reply: The changes in Siberian High can be seen more clearly from the sea level pressure (SLP) field (Figure R1). It is seen from the change in land-sea SLP gradient over East Asia-Pacific caused by El Niño (Figure R1 (a) and (c)) that El Niño causes

a weakness of EAWM. We also calculated the change in EAWM index (Li and Yang, 2010) caused by El Niño and La Niña, and also found that El Niño causes a weakness of EAWM. Please see the Figure S3 in the supplementary material.

[Figure]

Figure R1. Medians of the simulated differences in DJF sea level pressure between (a) EL and CLI, (b) LA and CLI, and (c) EL and LA, with black dots indicating that the differences between more than 70% pairs of ensemble members have the same sign with the median differences.

2. Line 5-8 page 9: Over northeastern China winter CONCsur of aerosols are decreased in both El Nino and La Nina, indicating that the decrease over northeastern China is not irrelevant to ENSO. So, the authors should describe this point carefully.

Reply: From the revised Figs. 9c and 9f, it seems that ENSO does not affects the atmospheric contents of aerosols over the north part of China (including northeastern China) as significantly as it does over southern China. As to this point, we have added a paragraph for discussion from the perspective of cold air frequency in China, please see section 4 in the revised manuscript.

3. Line 1 page 24: "zonal flux" should be "horizontal flux".

Reply: revised.

Interactive comment on "The Effects of El Niño-South Oscillation on the Winter Haze

Pollution of China" by Shuyun Zhao et al.

Anonymous Referee #3

**Using observations and model simulations, the authors investigated the possible influences of ENSO events on aerosols over China. Understanding the changes in aerosols is a relevant topic for improving our knowledge of relationship between natural cycle and aerosols. Based on the observational data, they concluded that the haze days of southern China tend to be less (more) than normal in El Niño (La Niña) winter, however the relationship is not significant. However, inconsistent result is seen based on the simulated result. And they concluded it is due to the fact that heavy haze days are more frequent in El Nino winter. I agree with the authors that the influence of ENSO events on aerosol concentrations over China cannot be ignored. And it is an important interdisciplinary issue which needs more attention and deep researches. However, there are many problems of this manuscript, and it cannot be accepted by ACP as it is now. Some specific comments or suggestions are listed as follows.**

1. The authors mainly focused on the influence of ENSO via EAWM on the haze over China, however, what is exact influences of the ENSO influence on the EAWM? And the involved physical processes. The authors did not give a detailed explanation regarding this point neither concluded the existed explanations. I suggest the authors do not avoid discussing this question, but make complete statement about this point.

Reply: Thank you for the suggestion. We have added discussions on how ENSO influences the EAWM in section 3.2.2. It was found that El Niño decreased the winter land-sea gradient in the sea level pressure field over East Asia and weakened the East Asian jet stream, therefore caused a weakness of EAWM. This is to some extent in accordance with previous studies. According to the work of Wang et al. (2000), the Pacific-East Asian teleconnection (PEAT) is the key bridge, which connects the ENSO

and its upstream climate in East Asia. Therefore, we discussed the PEAT from vorticity and stream function. Please see the revision and the Figures 3S and 4S in the supplementary document.

2. The influences of ENSO on the aerosols have been discussed in recent works. Regarding the impacts of El Niño and La Nina Modoki events on aerosol concentrations over eastern China, and the impacts of EASM on O3 over eastern China. The authors should include the relevant work into the present study, compare the differences, and further highlight the motivation of the present study.

Reply: We have cited more recent works involving the impacts of El Niño and La Niña Modoki events on aerosols over eastern China, and the impacts of EASM on $O_3$. More discussions have also added. Please see the section 1 of the revised manuscript.

3. The abbreviations of JJJ, JZH, and GG show no sense, the abbreviations should include geographic information and easy remember. I suggest using north China, southern China to replace.

Reply: replaced.

4. The contradict description in the manuscript, for example, P5L21, "After the year of 2000, the winter-average monthly haze days over the three regions grew dramatically, especially over JZH and GG"; but in P5L25, "haze days over JZH and GG increased too abruptly after the year of 2010, especially in 2013".

Reply: contradict description has been modified.

5. The reasons for the applying a linear trend removing and a 2-8 years band-pass filtering showed be illustrated, and the valid freedom of significant should be applied to the significance test. In Figure 3, it is seen that the hazes days show nonlinear variations, even within 1950-2010. And whether the hazes days show a 2-8 years' periods is not clear, it makes me hard to understand why the authors removed the linear trend and a band pass filter.

Reply: As has been pointed out by another reviewer that by applying a 2-8 years band-pass filtering, the linear trend had already been removed. Therefore, we have modified Figure 4, by only applying the 2-8 years band-pass filtering. The reason for applying the 2-8 years band-pass filtering is to remove the inter-decadal variabilities of both the winter haze days and SST.

We have illustrated the reason in the reviewed manuscript. Please see the section 3.1. The significance has been given when plotting the correlations between the winter haze days of China and SST, please see the figure caption of Figure 4.

We removed the linear trend before analyzing the relationship between the winter haze days of different regions of China and SST, because we wanted to remove the influences of global warming and the increasing anthropogenic emissions of aerosols and their precursors as much as possible. And we applied a 2-8 years band-pass filtering, in order to remove the inter-decadal variabilities, as it is known that ENSO is regulated by inter-decadal oscillations, e.g., PDO. And it can also be seen from Figure 3 that the winter haze days over northern and eastern China show an obvious peak around 1980, which also indicates an influence from some inter-decadal oscillation.

6. As to the influences of ENSO on the EAWM, many researchers have illustrated that the ENSO shows important role in impacting the intensity of EAWM, and the climate, including temperature, rainfall and winds over north China, central China are impacted by the EAWM. However, the result shows the role of ENSO on hazes over JJJ and JZH is not evident. I suggest the authors to separate the stronger and moderate ENSO events, or separate the eastern and central ENSO events to further explore the result, and it is unknown the exact years of El Nino and La Nina in the present work.

Reply: Inspired by the question, we rethought about this chain of ENSO–EAWM–haze pollution of the large north part of China. In January 2013, when the unprecedented haze event happened in Beijing, there was no El Niño or La Niña event in the tropical pacific. These reminded one of our authors that in her previous operational work she had noticed that El Niño did not necessarily reduce the cold air frequency in China in winter, even though it could weaken EAWM. In winter, a few clean days in the north part of China always follow a breakout of cold air from Siberia or Mongolia. The relationship between ENSO and the cold air frequency makes the connection between ENSO and the haze pollution of the north part of China more complex than it is expected from the perspective of the strength of EAWM. We have added a paragraph in the part of conclusions and discussions. Please see the revision.

In this work, we did not use the SST for any real specific year, but the climatology SST and a typical ENSO mode scaled by average Niño3.4 index, in order to diminish the influences from other forcings (e.g., in the Atlantic Ocean and Arctic) as much as possible. In our future work, we will change the shape of the inter-monthly variation of Niño3.4 (Figure 1b) and/or other

ENSO indices by selecting different El Niño and La Niña events or change the shape artificially, to explore the influence of ENSO of different types and strengths on the haze pollution over the large north part of China. Thanks for the suggestion.

7. The performance of the model used to perform the simulations in reproducing ENSO patterns as well as its variations is unknown. Since ENSO is a complex air-sea interaction process, it is necessary to access the performance in simulating ENSO whereas given the circulations over eastern China during winter and summer. And the observed variations in rainfall should be shown while the simulated result for the poor simulations of rainfall in the models.

Reply: In this work, we used an AGCM instead of CGCM, with prescribed SST of climatology and superimposed ENSO signals. It can only explore one-way (ENSO→atmosphere→aerosols) influences, as we have also pointed out in the part of conclusions and discussions. Now, we've added more discussions on the simulated circulation responses to ENSO, including the responses of East Asian jet stream, EAWM index, and the Pacific East Asian teleconnection. Please see the revised manuscript and supplementary document.

The observational relationship between ENSO and the rainfall over East Asia and the globe has been given on a website of the NCC/CMA (Figure R2 left). We added the citation of the website in the revision. Thanks for the suggestion.

[Figure]

Figure R2. left: Correlation coefficient between Nino3.4 and DJF precipitation given at http://cmdp.ncc-cma.net/pred/cn_enso.php?product=cn_enso_corr&season=DJF#corr; right: the revised figure 8c, differences in DJF precipitation in percentage between El Nino and La Nina winters.

8. Figure captions, the blank in figure 5a, b, should be mentioned, however not in c and d. Figure 7, shaded for hgt not contour. Figure 9, it is better if the relative changes is shown whereas the absolute values, considering that the climatology mean shows big differences as shown in Figures 1 and 6.

Reply: Captions of Figure 5 and 7 have been modified, please see the revised manuscript. Indeed, in Figure 6 the climatology mean aerosol concentrations and loadings show large geographical variations. However, over the areas east of 105ºE and south of 45ºN where we're interested in, their geographical variations are not very large. Therefore, we kept the absolute changes in Figure 9. Still very thank you for the suggestion.

**Part 2. A list of relevant changes made in the manuscript**

1. We replotted Figure 4, by only applying a band-pass filtering of 2-8 years, as band-pass filtering has already done the linear-trend removing, which is redundant in the original manuscript. Corresponding, the figure caption was also modified.

2. We replotted Figures 7-11, by adding the comparisons between El Niño and La Niña winters. Correspondingly, we added relevant illustrations in the figure captions and discussions in the text.

3. We added markers to Figures 7-11 where the differences between 70% pairs of experiments have the same sign with the median difference as a reflection of significance. Illustrations have also been added to the last paragraph of section 2.2.

4. We replotted Figure 12, by calculating the frequency distributions instead of the probability distribution functions of the winter daily surface concentrations (CONCsur), and we also added the number of days when the CONCsur $\leq 40\mu g/m^3$, $40 < $ CONCsur $\leq 80\mu g/m^3$, and CONCsur $> 80\mu g/m^3$. Caption of Figure 12 was also modified. In the text section 3.2.3 and abstract, we also modified relevant discussions.

5. We added 5 more references, e.g., Feng et al. (2017), Kang and Lee (2017), Li and Yang (2010), Wu et al. (2013), and Yang et al. (2014) in the text and reference list.

6. We added a figure in the supplementary material (Figure S1) to illustrate the locations of the provinces and regions of China

mentioned in section 3.1.

7. We added the comparisons as to the winter haze days over southern China between 4 pairs of El Niño and La Niña winters in the supplementary material (Figure S2). And relevant discussions were also added in the text, e.g., section 3.1.

8. We modified some paragraphs in section 3.2.2, and added extensive discussions about how ENSO affected the East Asian Winter Monsoon (EAWM). We added a citation of the website of NCC/CMA (http://cmdp.ncc-cma.net/pred/cn_enso.php?product=cn_enso_corr&season=DJF#corr) and two figures in the supplementary material (Figures S3 and S4) to illustrate how ENSO affected the EAWM from the perspective of East Asian jet stream and Pacific-East Asian teleconnection.

9. We added a paragraph in section 4 (conclusions and discussions) to illustrate why the connections between ENSO and the haze pollution over northern and eastern China are not as strong as we expected to be, as they both have some connections with the EAWM. During illustration, we added two figures in the supplementary material (Figures S5 and S6).

10. Other modifications relative to grammar corrections or making some sentences more clearly and properly.

**Part 3. Marked-up manuscript version**

[revised manuscript text omitted]

---

## Author Response (AR2)

**Part 1. Point-by-point response to the reviewers**

**Responses to reviewers' comments and suggestions**

We greatly appreciate all the reviewers' comments and suggestions, which are very important in helping us to improve the manuscript. Our detail responses are below in blue.

**Anonymous Referee #1**

**For final publication, the manuscript should be accepted as is subject to minor revisions.**

**Suggestions for revision or reasons for rejection**

The revised manuscript gives more detailed explanations compared to the previous one, so I think it is easier to read. On the other hand, I have one comment for the contexture of the presentation. The major finding about different frequency distribution of CONCsur over the southern China for two ENSO phases has been presented in the last figure (Fig. 12). I recommend that the important figure, Fig.12 is represented at earlier part of the manuscript (e.g., after Fig. 9) to show the inverse feature for the haze days and atmospheric contents of aerosols during the ENSO phase. If you explain the reasons in Figs. 7 ~ 11 after Fig. 12, I feel it helps the reader easier to understand the question about the inverse relationship. However, I leave it to the author's decision.

Reply: Fig.12 has been moved forward after Fig. 9. Therefore, figure captions and relevant descriptions in the text were also modified. Thanks for the suggestion.

**Anonymous Referee #2**

**For final publication, the manuscript should be accepted as is subject to minor revisions.**

**Suggestions for revision or reasons for rejection**

In revised version, the authors faithfully answered reviewer' suggestions including new additional figures and discussions. However, one thing I would like to point out is the level of significance of the response. This study investigates the impact on ENSO forcing, and this ENSO forcing is already a response experiment for extreme SST

forcing. So, the response of climate system on the El Nino and La Nina forcing can be extreme in different directions. In addition, we think that 20 pieces of sample each are sufficient in estimating the significance of the difference between two groups. However, the authors present a median difference for both responses instead of a significance level, and also provide a percentage (70%) of ensemble members with the same sign as the median difference. This method can show to some extent that there is a difference between the two groups, but it is hardly the best method. Therefore, it is recommended to present the significance level of the difference between the two groups to the supplementary material at the level of 5%, 10%, 20%, and so on. This information can provide readers with further insight at regional scale on the effects of ENSO on the winter haze pollution of China.

Reply: Additional figure presenting the differences in winter-average aerosol surface concentrations and loadings between the groups of EL and LA were added in the supplementary material (Figure S5), with different significance levels. Thanks for the suggestion.

**Anonymous Referee #3**

**For final publication, the manuscript should be accepted subject to technical corrections.**

**Suggestions for revision or reasons for rejection**

No.

Reply: We have checked the manuscript again about grammar, figures, captions, and references. Thanks for reviewing our work.

**Part 2. A list of relevant changes made in the manuscript**

1. According to the suggestion of reviewer, we have moved Figure 12 forward after Figure 9, therefore, figure captions and relevant descriptions were also modified.

2. We have added a figure to the supplementary material showing the mean differences in winter surface aerosol

concentrations and loadings between EL and LA, with different significance levels.

3. Other modifications as to grammar problems (mainly about the tense), references, etc.

**Part 3. Marked-up manuscript version**

[revised manuscript text omitted]